# *Mesocricetus auratus* (Golden Syrian Hamster) Experimental Model of SARS-CoV-2 Infection Reveals That Lung Injury Is Associated with Phenotypic Differences Between SARS-CoV-2 Variants

**DOI:** 10.3390/v17081048

**Published:** 2025-07-28

**Authors:** Daniela del Rosario Flores Rodrigues, Alexandre dos Santos da Silva, Arthur Daniel Rocha Alves, Bárbara Araujo Rossi, Richard de Almeida Lima, Sarah Beatriz Salvador Castro Faria, Oswaldo Gonçalves Cruz, Rodrigo Muller, Julio Scharfstein, Amanda Roberta Revoredo Vicentino, Aline da Rocha Matos, João Paulo Rodrigues dos Santos, Pedro Paulo Abreu Manso, Milla Bezerra Paiva, Debora Ferreira Barreto-Vieira, Gabriela Cardoso Caldas, Marcelo Pelajo Machado, Marcelo Alves Pinto

**Affiliations:** 1Laboratory of Technological Development in Virology, Oswaldo Cruz Institute, Oswaldo Cruz Foundation, Rio de Janeiro 21040-900, Brazil; alexsantos@ioc.fiocruz.br (A.d.S.d.S.); art-dan@hotmail.com (A.D.R.A.); barbararossi@aluno.fiocruz.br (B.A.R.); richard.lima@ioc.fiocruz.br (R.d.A.L.); sarah.faria@aluno.fiocruz.br (S.B.S.C.F.); 2Scientific Computing Program, Oswaldo Cruz Foundation, Rio de Janeiro 21040-900, Brazil; oswaldo.cruz@fiocruz.br; 3Animal Experimentation Laboratory, Institute of Technology in Immunobiologicals, Bio-Manguinhos, Oswaldo Cruz Foundation, Rio de Janeiro 21040-900, Brazil; rmuller@bio.fiocruz.br; 4Laboratory of Molecular and Cellular Immunology, Federal University of Rio de Janeiro, Rio de Janeiro 21941-599, Brazil; jscharf2@gmail.com (J.S.); amandarrvicentino@gmail.com (A.R.R.V.); 5Laboratory of Respiratory Viruses, Exanthematics, Enteroviruses and Viral Emergencies, Oswaldo Cruz Institute, Oswaldo Cruz Foundation, Rio de Janeiro 21040-900, Brazil; aline.matos@ioc.fiocruz.br; 6Experimental Medicine and Health Laboratory, Oswaldo Cruz Institute, Oswaldo Cruz Foundation, Rio de Janeiro 21040-900, Brazil; joaopaulorsantos@gmail.com (J.P.R.d.S.); mansoppa@gmail.com (P.P.A.M.); millabpaiva@gmail.com (M.B.P.); marcelo.pelajo@fiocruz.br (M.P.M.); 7Laboratory of Viral Morphology and Morphogenesis, Oswaldo Cruz Institute, Oswaldo Cruz Foundation, Rio de Janeiro 21040-900, Brazil; barreto@ioc.fiocruz.br (D.F.B.-V.); gabrielacardosocaldas@gmail.com (G.C.C.)

**Keywords:** SARS-CoV-2, golden Syrian hamster, compensatory mechanisms, innate immune response, pathogenesis, endotheliitis, podoplanin, HMGB-1, animal model

## Abstract

Despite the current level of public immunity to SARS-CoV-2, the early inflammatory events associated with respiratory distress in COVID-19 patients are not fully elucidated. Syrian golden hamsters, facultative hibernators, recapitulate the phenotype of SARS-CoV-2-induced severe acute respiratory syndrome coronavirus 2 (SARS-CoV-2)—induced severe acute lung injury seen in patients. In this study, we describe the predominance of the innate immune response in hamsters inoculated with four different SARS-CoV-2 variants, underscoring phenotypic differences among them. Severe inflammatory lung injury was chronologically associated with acute and significant weight loss, mainly in animals inoculated with A.2 and Delta variants. Omicron-infected animals had lower overall histopathology scores compared to other variants. We highlight the central role of endothelial injury and activation in the pathogenesis of experimental SARS-CoV-2 infection in hamsters, characterised by the presence of proliferative type I and type II pneumocytes with abundant surfactant expression, thereby maintaining hyperinflated alveolar fields. Additionally, there was evidence of intrapulmonary lymphatic vessel proliferation, which was accompanied by a lack of detectable microthrombosis in the lung parenchyma. However, white microthrombi were observed in lymphatic vessels. Our findings suggest that the physiological compensatory mechanisms that maintain respiratory homeostasis in Golden Syrian hamsters prevent severe respiratory distress and death after SARS-CoV-2 infection.

## 1. Introduction

The coronavirus disease 2019 (COVID-19) pandemic, caused by severe acute respiratory syndrome coronavirus 2 (SARS-CoV-2), has been significantly impacted by the rapid evolution of the virus and the emergence of variants responsible for millions of severe human cases worldwide. These variants are characterised mainly by their differential virulence, as shown by their ability to spread in the environment, to be transmitted, and to infect [1,2,3].

Despite the end of the COVID-19 pandemic, the mechanisms of virus-induced acute lung injury and an extensive range of clinical severity signs remain unclear. Many relevant hypotheses have already been evaluated, such as the presence of some comorbidities, such as obesity and diabetes, that are associated with the worst outcomes due to compromised innate and humoral immunity [4]. In addition, virus infection has been shown to induce cytopathic effects on type 2 pneumocytes (AT2) [5], reduce antiviral interferon type I expression in the respiratory tract [6,7], and dysregulate the renin-angiotensin system, which reduces pulmonary perfusion and leads to parenchymal necrosis and the formation of microthrombi via endothelial cell injury and shear stress via hyperperfusion of collapsed lung tissues in patients [8,9]. Thus, the innate components of the immune system play a crucial role in the severe outcome of COVID-19.

In addition to humans and nonhuman primates, any animal species that naturally expresses ACE-2 with some homology to the corresponding human receptor could maintain variable levels of SARS-CoV-2 replication in the respiratory tract. Hamsters (Golden Syrian and Roborovski) and human ACE2 transgenic mice are susceptible to SARS-CoV-2 in laboratory-controlled conditions [10,11]. Transgenic mice expressing human ACE2 (hACE-2) [12], K18 mice, presented discrepant mortality rates, with lethal “unexpected” viral neurotropism [13], suggesting an aberrant systemic distribution of hACE-2.

However, *Mesocricetus auratus*, a small laboratory animal rodent, recapitulates the phenotype of SARS-CoV-2-induced acute respiratory syndrome coronavirus 2 (SARS-CoV-2), detected in patients. Despite the differences in immunology and respiratory anatomy between humans and hamsters [14], they are the fifth most commonly used small animal in biomedical research and constitute a particularly useful model for lung disease [15] that naturally resembles a wide range of COVID-19 manifestations in patients [13,16]. Its susceptibility is justified by the elevated homology (88,89%) between human and hamster ACE-2 [17], and experimental data have demonstrated that lung infection is restricted to sites containing both ACE2 and TMPRSS2, both of which are colocalised in tertiary bronchi, bronchioles, and alveoli [18,19]. Thus, the hamster was susceptible enough to be successfully challenged with distinct SARS-CoV-2 variants [20,21].

Importantly, the Syrian hamster (*Mesocricetus auratus*) is a facultative hibernator; the physiological mechanisms underlying this facultative hibernation could offer valuable insights for developing strategies to protect human organs, particularly in the context of viral infections [22]. However, one caveat of hamster models is the lack of research tools for this species—they remain scarce when compared (for example) with those available for mice [10].

In our study, we adopted the Syrian hamster as an experimental animal model for SARS-CoV-2 infection due to its susceptibility to natural lung disease. The phenotypes of the animals intranasally inoculated with four different variants of SARS-CoV-2 were clinically, morphologically, and virologically evaluated. Both cytopathic and innate immune components induced by SARS-CoV-2 variant infection were identified in the hamster respiratory tract, with endothelial cell injury playing a central role in disease immunopathology. Previous studies have also conducted comparative analyses by inoculating Syrian hamsters with different SARS-CoV-2 variants, identifying body weight loss as a reliable clinical indicator of infection, showing that the virus induces pathological features resembling bronchopneumonia observed in COVID-19 patients, and that animals infected with the Omicron variant exhibit milder pneumonia compared to those infected with other variants [23,24,25,26]. The present study distinguishes itself by investigating the pathophysiology of this viral infection in light of the hamster’s unique status as a facultative hibernator. This physiological trait is central to our approach and provides a novel perspective for understanding this host response.

## 2. Materials and Methods

### 2.1. Ethical Aspects

Our experimental research protocol was previously approved by the Ethics Committee for the Use of Laboratory Animals of Fiocruz (CEUA LW-9/20), and it was carried out following the Fiocruz Guide for the Care and Use of Laboratory Animals and the CONCEA/MCT guidelines (https://www.gov.br/mcti/pt-br/composicao/conselhos/concea (accessed on 30 June 2020)) and ARRIVE guidelines 2.0 [27].

### 2.2. SARS-CoV-2 Inocula

SARS-CoV-2 inocula were previously isolated from nasopharyngeal and/or throat swabs from COVID-19 patients from Brazil and kindly provided by Laboratório de Vírus Respiratórios, Exantemáticas, Enterovírus e Emergências Virais (Fiocruz-IOC, Rio de Janeiro, Brazil), Laboratório de Tecnologia Virológica (Fiocruz-Biomanguinhos, Rio de Janeiro, Brazil) and Laboratório de Virologia Molecular (UFRJ, Rio de Janeiro, Brazil). Isolates were propagated and amplified in the Vero E6, Vero hSLAM or VerohACE2/hTMPRSS2 cell lines and sequenced. Virus infection in the animals was performed with A.2 (GISAID: EPI_ISL_528539), Gamma (GISAID: EPI_ISL_18433590), Delta (GISAID: EPI_ISL_2645417) or Omicron BA.1 (GISAID: EPI_ISL_8430488) variants. The variants were propagated and amplified in the Vero E6 cell line to form the inoculum.

### 2.3. SARS-CoV-2 Infection in Golden Syrian Hamsters

Healthy male and female golden Syrian hamsters (*Mesocricetus auratus*) adults were obtained from the Bahia Fiocruz (Bahia, Brazil) breeding colony and housed in an animal biosafety level 3 (ABSL3) facility (Laboratorio de Experimentação Animal—LAEAN/Bio-Manguinhos/Fiocruz, Rio de Janeiro, Brasil). The LAEAN staff was previously qualified to work in ABSL3. During the experimental period, the animals were housed in microisolators and stainless-steel racks (ALESCO, São Paulo, Brazil) in climate-controlled rooms (temperature of 22 ± 1 °C and humidity of 55 ± 5%) with a 12 h light/dark cycle and were fed a commercial hamster diet with water provided ad libitum. Before starting the study, they were kept in microisolators during the quarantine period for adaptation to the ABSL3 facility, during which their preinoculation baseline parameters were recorded individually.

The animals were divided into 5 groups, with 3 animals per microisolator. One group was inoculated intranasally with A.2 (*n* = 24), the second group was inoculated with Gamma P.1 (*n* = 12), the third group was inoculated with the Delta AY.6 variant (*n* = 12), the fourth group was inoculated with the Omicron BA.1 variant (*n* = 12), and the last group was the negative control group inoculated with PBS (12 animals). To reproduce SARS-CoV-2 infection, all hamsters were manipulated in a currently validated laminar flow in the ABSL3 facility. The animals were infected intranasally with 50 µL of the SARS-CoV-2 isolate, corresponding to 10^6 plaque-forming units (PFU)/mL. The negative control group was inoculated intranasally with 50 µL of PBS.

The golden Syrian hamsters were followed for 15 days post inoculation (DPI). The physical examination results, clinical manifestations and body weights were measured daily. At 3, 5, 10, and 15 DPI, three animals from each group were randomly euthanised. Euthanasia procedures were performed by total exsanguination, with blood drained by heart puncture, under deep surgical anaesthesia that consisted of ketamine hydrochloride at 50 mg/kg (Vetanarcol, König, Argentina), xylazine hydrochloride at 0.1 mg/kg (Syntec Brazil, São Paulo, Brazil) and midazolam hydrochloride at 1.0 mg/kg (Roche, Farmacêutica do Brasil, São Paulo, Brazil). For deep anaesthesia, 2.5% sodium thiopental at 90 mg/kg (Thiopentax, Cristalia, São Paulo, Brazil) was delivered intravenously.

Furthermore, necropsies were performed, and sera, oropharyngeal swabs, and tissue samples (lung, brain, liver and intestine) were collected at the euthanasia corresponding to the DPI. The tissue samples were homogenised in PBS with a tissue Ruptor (Qiagen, Hilden, Germany), followed by centrifugation (20 min, 3000 rpm). All of the samples were stored at −70 °C until analysis. Blood obtained by total exsanguination was used to evaluate the neutralisation rate, biochemical markers (glucose, total bilirubin and fractions, creatinine, albumin, total cholesterol, alanine aminotransferase, and aspartate aminotransferase), and blood count tests (leukocytes, bands, segmented neutrophils, eosinophils, lymphocytes, monocytes, and platelets). Oropharyngeal swabs and tissues were used to evaluate the presence of SARS-CoV-2 RNA via RT—PCR. Additionally, lung samples were used to assess histopathology and SARS-CoV-2 antigen detection in the respiratory tract.

### 2.4. SARS-CoV-2 Neutralising Antibodies Detection in Blood Samples

To detect neutralising antibodies against SARS-CoV-2 after hamster infection, blood samples were collected from each animal under deep anaesthesia during cardiac puncture. The neutralising antibody titres were assessed via the commercial cPass SARS-CoV-2 Neutralisation Antibody Detection Kit (catalogue number L00847, GenScript USA Inc./Nanjing GenScript Diagnostics Technology Co., Ltd. Nanjing, China). The methodological procedures followed the product manufacturer’s instructions (Version 7.0, Update 2022.02.01), and antibody neutralisation titres above 30% were considered neutralising for SARS-CoV-2.

### 2.5. Quantitative SARS-CoV-2 RNA Detection in Oropharyngeal Swabs and Tissue Samples

We investigated the distribution of SARS-CoV-2 RNA in the animals by analysing the oropharyngeal swabs and tissues that were collected during necropsies. The genetic material of the samples was extracted and purified via the RNA/DNA 300 kit H96 in the Janus G3 and Janus Chemagic automatic extractor (Perkin-Elmer, Waltham, MA, USA) in accordance with the manufacturer’s instructions. For SARS-CoV-2 detection, a molecular kit with primers and probes for the envelope (E) gene region was used (Bio-Manguinhos, Rio de Janeiro, Brazil) following the manufacturer’s instructions. Viral genomic RNA quantification was performed using an in-house ssRNA standard curve with a previously quantified sample. Samples with a cycle threshold (CT) value lower than 37 for the SARS-CoV-2 E target were considered positive.

### 2.6. Inflammatory Markers Quantification

Myeloperoxidase, cysteinyl leukotriene (Cys-leukotriene) and C-reactive protein (CRP) levels were quantified as inflammatory markers in the infected animals. Myeloperoxidase levels were investigated with a commercial quantitative hamster myeloperoxidase ELISA kit (MBS026673, MyBiosource) for the quantification of biological neutrophil reactivity in the animal lung parenchyma. Cysteinyl leukotriene levels were measured with an ELISA kit (Cayman Chemical, Ann Arbor, MI, USA, Item No. 500390) and were used to quantify the leukocyte reactivity in the lung parenchyma. Serum samples collected during the total exsanguination (euthanasia step) were utilised to analyse CRP levels, as they increase in response to inflammation, via a quantitative hamster-specific ELISA (MyBiosource, MBS045692).

### 2.7. Immunofluorescence

Immunofluorescence was performed to evaluate the levels of podoplanin, pan-cytokeratin and high-mobility group box 1 (HMGB1) antigen in the lung samples. For this purpose, paraffin-embedded lung sections were dewaxed and hydrated. Further antigen retrieval was performed by heating the sections in 10 mM sodium citrate buffer pH 6.0 (Merck, Darmstadt, Germany) for 15 min at 120 °C. Primary antibodies were diluted in PBS, applied to the sections and incubated overnight at 4 °C in a humid chamber. The primary antibodies used were as follows: anti-SARS-CoV-2 nucleocapsid protein (Thermo Fisher, Waltham, MA, USA, cat. PA1-41098, 1:500); anti-hamster podoplanin anti-281-mG2a-F (produced and provided by Dr. Yukinari Kato, Tohoku University Graduate School of Medicine-Japan, 1:1000); mouse anti-human pan-cytokeratin (Thermo Fisher, MA5-13156, 1:200); and polyclonal anti-HMGB1 (Abcam, Cambridge, UK, Ab18256, 1:100). The secondary antibodies were applied to the sections and incubated for 1 h at 37 °C. The secondary antibodies used were as follows: goat anti-rabbit IgG (H + L) cross-adsorbed secondary antibody, Alexa Fluor™ 488 (Invitrogen, 1:750) and goat anti-mouse IgG (H + L) cross-adsorbed secondary antibody (Invitrogen, 1:750). Nuclear staining was performed with 4,6-diamidine-2′-phenylindole dihydrochloride (DAPI; Roche, Basel, Switzerland; 1:5000) for 10 min at room temperature.

The sections were counterstained with Evans blue (Sigma-Aldrich, St. Louis, MS, USA; 1:10,000) for 10 s. ProLong Gold (Invitrogen, Waltham, MA, USA, P36930) was used as an antifade mounting medium for the coverslips. The samples were observed with a Zeiss Colibri and/or a confocal LSM 710 (Fiocruz platform RPT07A). The fluorescence of specific regions was detected using a Zeiss Elyra PS.1 Superresolution microscope. As a negative control, lung sections were incubated with PBS instead of a monoclonal antibody, followed by incubation with a secondary anti-rabbit antibody for all analyses. The images of the fluorescent fields were quantified via the ImageJ 1.53e programme (https://imagej.nih.gov (accessed on 30 September 2021)).

### 2.8. Histopathology and Transmission Electron Microscopy

Lung samples stored in 10% modified Carson-Millonig formalin pH 7.0 (Sigma-Aldrich, St. Louis, MO, USA) were embedded in paraffin according to standard methods. The paraffin blocks were sectioned at 4 μm and stained with haematoxylin—eosin. For the pulmonary histomorphological analysis, a detailed microscopic description of the slides was employed, and a semiquantitative assessment was proposed using scores that encompass different microscopic characteristics. These characteristics are widely used in experimental models for acute interstitial pneumonia [28,29,30,31,32]. In this study, we established five scores (0, 1, 2, 3, and 4) for histopathological alterations according to the intensity of the alterations. These include the distribution patterns of the alterations, the inflammatory infiltrate, vascular alterations, and other findings, which were analysed via an AxioScope brightfield microscope and Axiocam HRc digital camera.

For ultrastructural analysis via transmission electron microscopy (TEM), samples of Syrian hamster lungs infected with SARS-CoV-2 (Delta and Ômicron) were collected at 3 and 10 DPI and fixed by immersion in 2.5% glutaraldehyde in 0.1 M sodium cacodylate buffer, pH 7.2 (Electron Microscopy Sciences, Hatfield, PA, USA). The samples were subsequently fixed in 1% buffered osmium tetroxide (Electron Microscopy Sciences, Hatfield, USA), dehydrated in acetone, embedded in epoxy resin (Electron Microscopy Sciences, Hatfield, USA), and polymerized at 60 °C for three days [33]. Ultrathin sections (50–70 nm) were obtained from the resin blocks. Lung samples from uninfected animals were used as controls and processed under the same conditions. The sections were placed on copper grids (300 mesh and no coating) and observed under a Hitachi HT 7800 (Hitachi, Tokyo, Japan) transmission electron microscope.

### 2.9. Statistical Analysis

All analyses were performed using the R Project for Statistical Computing programme (http://www.r-project.org/ (accessed on 31 May 2024)). Regression analysis was carried out using a general linear model (GLM) to estimate the variation in a dependent variable in terms of a combination of several reference functions. The obtained beta estimate values were used to predict alterations. Gamma, Delta, and Omicron variant findings were compared with those of the A.2 variant. Two-tailed Kruskal-Wallis tests and Student’s *t* tests were used for comparative analysis, and *p* < 0.05 was considered statistically significant in our study.

## 3. Results

### 3.1. Clinical Manifestations

All animals were clinically followed daily and weighed in the BSLA3 facility. No clinical signs of respiratory distress, shaking chills, lethargy, or anorexia were observed during the study period. The body temperature data were assessed by rectal temperature, but the data were discarded in this study because the procedure produced intense stress in response to the manipulation, with consequent body temperature elevation.

To carry out a sophisticated statistical analysis and, at the same time, circumvent the limitations imposed by the pandemic, such as the small and decreasing number of animals in each group, we applied GLM, a standard regression model, to analyse the probability of the response variable occurring considering the potential contribution of the explanatory (predictor) variables.

We used Weight ~ dpi + variant to analyse whether days post inoculation and the inoculated variant are significant factors for the weight response variable. Negative estimated β (beta) values, with a confidence interval that does not include zero, indicate the potential contribution to weight loss (response variable) of the predictor variables (DPI and variants) that showed statistically significant *p*-values.

The application of GLM revealed that the weight loss of the animals was related to the days post inoculation (DPIs) and the SARS-CoV-2 variant inoculated. The interaction of the period between +2 and +7 dpi and the variants A.2 and Delta showed a statistically significant factor for the average weight loss, as represented by the beta values shown in Figure 1.

### 3.2. Blood Biochemistry and Haematological Findings

In this work, the following biochemistry markers were evaluated: glucose, total bilirubin and fractions, creatinine, albumin, total cholesterol, alanine aminotransferase (ALT/TGP) and aspartate aminotransferase (AST/TGO). The baseline levels of biochemistry markers were the average levels in the negative control group. During the 15 days of SARS-CoV-2 infection, some biochemical changes were detected in the blood. Our study’s blood markers of liver function were ALT/TGP, AST/TGO, albumin, total bilirubin and fraction levels.

No statistically significant differences in the ALT/TGP ratio were found among the infected groups. The application of GLM revealed that inoculation with the Gamma variant contributed to elevated AST/TGO ratios (beta estimate: 52.62, CI: 12.74–92.5, *p* = 0.010). Although statistically significant differences in AST/TGO biochemical markers were not detected for the A.2, Delta and Omicron variants, elevated levels of AST/TGO were detected on 15 DPI in 1 animal from the group infected with the A.2 variant (4 times greater than the baseline level) and Delta (2 times greater than the baseline level). Additionally, 2 animals from the group infected with the Omicron variant presented elevated levels (2 times above the baseline level) of AST/TGO on 10 DPI. The albumin and bilirubin levels did not change during the study period.

To confirm normal kidney function, the blood creatinine and total cholesterol levels were monitored during the study, and they were maintained unchanged. The GLM analysis revealed a tendency towards a decrease in glucose levels in Gamma-inoculated animals (beta estimate: −32.04, *CI*: −49.32–−14.76, *p* < 0.001).

In our study, hyperglycaemia was suggested for 10 inoculated hamsters (>150 mg/dL), and elevated levels of glycated haemoglobin (HbA1c) were detected in one animal, both at the exudative step of SARS-CoV-2 infection in a variant-inoculated independent manner. However, we assume technical limitations in our experimental design, such as the absence of a previous fasting period for the animals.

Thus, with these results, we cannot confirm that the hamster model spontaneously reproduces the metabolic disorders described in humans.

Concerning the haematological findings, elevated eosinophil counts in the blood were detected mainly in A.2-inoculated hamsters (3 and 5 DPI) and moderate thrombocytopenia (<300.000 platelets/mL) was observed in A.2 (3, 5, and 10 DPI), Delta (10 DPI) and Omicron (10 and 15 DPI) inoculated hamsters.

### 3.3. Production of Neutralisation Antibodies

Early production of neutralising antibodies at 5 DPI was detected only in animals infected with the Gamma and Delta variants (30.18% and 54.81%, respectively). Neutralising antibodies appeared at 10 DPI in animals infected with A.2 and Omicron (86.75% and 60.86%, respectively). The lowest level of neutralising antibodies was observed in the Omicron variant group (Table 1).

### 3.4. Quantitative SARS-CoV-2 RNA Detection in Swab and Tissue Samples

Oropharyngeal tract and tissue samples were collected during the euthanasia. Each variant presented different SARS-CoV-2 RNA contents throughout the course of infection, as depicted in Figure 2.

The highest average level of SARS-CoV-2 RNA found in the oropharyngeal tract (Figure 2A) was detected in the A.2-inoculated animals at 3 DPI (10^8.25^ SARS-CoV-2 RNA copies). When the lungs were evaluated (Figure 2B), the highest average level of SARS-CoV-2 RNA was detected in the Delta variant group at 5 DPI (10^8.11^ SARS-CoV-2 RNA copies). The highest average level of SARS-CoV-2 RNA in the liver parenchyma (Figure 2C) was found in the Delta variant group at 5 DPI (10^4.26^ SARS-CoV-2 RNA copies). The brain samples of the animals infected with the Gamma variant at 3 DPI presented the highest viral load (10^3.01^ SARS-CoV-2 RNA copies), as represented in Figure 2D, and the highest intestinal viral load (Figure 2E) was detected in the animals infected with the Delta variant at 3 DPI (10^3.43^ SARS-CoV-2 RNA copies). Syrian hamster oropharyngeal tract and lung samples were positive for SARS-CoV-2 RNA on all days investigated (Figure 2A,B). Liver and intestine (Figure 2C,E) samples presented undetectable viral RNA from 10 DPI onwards. The SARS-CoV-2 RNA content in brain samples was detected only in one hamster inoculated with A.2 (Figure 2D) at 15 DPI (10^1.12^ SARS-CoV-2 RNA copies).

The GLM analysis used the A.2 variant and 3 DPI as reference factors to evaluate the SARS-CoV-2 RNA content of all of the organs collected (Table 2). The number of viral RNA copies in the oropharyngeal tract decreased at 10 and 15 DPI for all of the variants. Delta and Omicron inoculation were associated with a decreased viral load in the upper respiratory tract. In the lungs, the viral load increased at 5 days post-infection (DPI). Still, it decreased at 10 and 15 DPI, with Delta being a factor associated with an elevated viral load and Omicron being a factor associated with a decreased viral load. The liver parenchyma showed a reduction in the viral load at 10 and 15 DPI, and Gamma, Delta and Omicron inoculations were risk factors for an increase in the viral load. Brain and intestinal samples from animals infected with SARS-CoV-2 presented a reduction in the viral load after 3 DPI. In addition, Gamma, Delta and Omicron variants were risk factors for increased viral load in the intestines of animals infected with SARS-CoV-2.

Despite the detection of viral RNA in tissues other than the oropharyngeal tract and the lung, such as the brain, liver and intestine, tissue lesions were only observed in the respiratory tract, confirming the respiratory tropism of SARS-CoV-2 in this hamster model.

### 3.5. Inflammatory Markers

The inflammatory markers myeloperoxidase, cys-leukotriene and C-reactive protein were evaluated in Syrian hamsters infected with SARS-CoV-2 and compared with those in the negative control group. There were no differences between the groups for the levels of cys-leukotriene, C-reactive protein and myeloperoxidase.

### 3.6. Immunofluorescence Reaction for SARS-CoV-2 Nucleocapsid, Lymphatic Vessel Endothelial Marker (Podoplanin) and HMGB1 Translocation in Lungs of SARS-CoV-2 Inoculated Hamsters

Initially, there was positive immunolabelling for SARS-CoV-2 antigens in the monolayer of the upper respiratory epithelium, mainly the trachea and bronchi, with the reactivity of type 1 pneumocytes. Subsequently, the epithelial cells detached from the bronchial epithelium (positive for the viral nucleocapsid) fall into the airway lumen and migrate to the respiratory epithelium (Figure 3A), as proven by the presence of immunolabelled viral particles inside alveolar epithelial cells (Figure 3B).

Immunolabelling for cytokeratin confirmed the epithelial origin of the detached intrabronchial cells and highlighted hyperplasia of the upper and lower airway epithelium (Figure 3C). In addition to hyperplasia, abnormal cells with atypia (cariomegaly, cellular pleomorphism, and prominent nucleoli) were confirmed to be of epithelial origin (Figure 3D).

Podoplanin-positive lymphatic endothelial cells exhibited ectasia in the presence or absence of intraluminal leukocytes (Figure 3E). It was also possible to observe the location of some lymphatic vessels adjacent to and around the circumference of the bronchial arteries (Figure 3F). Type I pneumocytes were also positive for podoplanin, and the absence of fluorescence in cells of the vascular endothelium was also observed.

To demonstrate the role of the innate immune response in SARS-CoV-2-induced lung injury, HMGB1 translocation was assessed. As shown in Figure 4, immunofluorescence staining revealed the presence of HMGB1, mainly in pneumocyte type 2 cells, with a few cells showing cytoplasmic HMGB1 staining for both the A.2 and Gamma variants at 5 and 10 DPI (Figure 4A, insets). We investigated whether infection could lead to different redox forms of HMGB1. The alarmin HMGB1 is susceptible to intracellular redox modifications, resulting in an all-thiol form, a disulfide form (Cys23-Cys45 intramolecular disulfide bond with the Cys106 thiol form), or a fully oxidised form. Under normal conditions, HMGB1 inside the nucleus is fully reduced, as previously reported [34]. In the injured lungs of the animals infected with both the A.2 and Gamma variants (compared with noninfected animals (NI) which showed a predominant upper band, light grey), Western blot analysis revealed the appearance of a lower band (dark grey bar). The appearance of the lower band suggested the formation of the disulfide form of HMGB1 (dsHMGB1). Our qualitative data suggest that the presence of dsHMGB1 is increased at 5 DPI by the Gamma variant. However, for A.2 variant infection, it seems that the dsHMGB1 form persisted abundantly until 10 DPI (Figure 4B).

### 3.7. Morphological Changes Induced in the Respiratory Tract of Syrian Hamsters After Experimental SARS-CoV-2 Inoculation

After being euthanized, the animals were necropsied, and the respiratory tracts were collected individually. Macroscopically, hepatized well-delimited areas were easily observed bilaterally between 3 and 10 DPI (Figure 5A). These areas were surrounded by normal delicately pink areas, interpreted as consolidated areas of the lung parenchyma, which were confirmed by histopathological analysis, contributing to defining segmentary pneumonia as a definitive diagnostic for all inoculated animals, without predominance between the right and left lungs, as well as the sex or age of the animals. At 10 and 15 DPI, we detected a progressive reduction in hepatized pulmonary areas and redness.

For all variants that exhibited SARS-CoV-2-induced inflammatory lung injury in the inoculated hamsters 72 h after intranasal inoculation, the earliest microscopic aspect of the lung injury observed was segmental interstitial pneumonia with strong mononuclear inflammation, alveolar oedema, and a random distribution of hepatized areas throughout the lung, without any lung-side predominance. For comprehensive microscopic analysis, we explored respiratory SARS-CoV-2-induced injury in the airway, alveolar, and vascular compartments. At 3 to 5 DPI, the animals presented with epithelial congestion, epithelial upper airway and bronchial desquamation with cilia loss due to subepithelial mononuclear infiltration with rupture of the lamina propria (Figure 5B,D) and submucosa layers and the presence of megakaryocytes and eosinophils (highlighted in Figure 5C), with haemorrhagic areas. The detached epithelial cells were often associated with the formation of epithelial plugs in the deep bronchial lumen and with alveolar collapse (Figure 5D).

In the respiratory vascular compartment, vasculitis/endotheliitis was a frequent feature, with the evident presence of activated lymphoblasts, subendothelial mononuclear infiltration, degeneration and oedema of the vascular wall (Figure 5E). These findings may precede the formation of fistulas and haemorrhage (Figure 5G). Atypical epithelial hyperplasia was frequently observed from 5 DPI until 15 DPI. Mitotic figures, karyomegaly, anisocytosis, prominent nucleoli and cellular pleomorphism associated with interstitial inflammatory cells were observed (Figure 5F). The presence of a hyaline membrane was rare on the alveolar surface. Diffuse alveolar injury was frequently observed at 5 DPI. The regenerative aspect of SARS-CoV-2 injury may be confirmed by epithelial cell regeneration at 10 DPI. At the recovery step, hyperinflated areas were associated with a reduction in infiltrated lymphocytes/lymphoblastoid cells and macrophages in the lung interstitial space. Despite the reduction in lung parenchyma inflammation, the metaplasia of respiratory epithelia and stromal fistulae to the bronchial lumen were also detected at 15 DPI. Lymphangitis associated with lymphangiectasia was most evident around the bronchial hilum (Figure 5C,H). Microthrombi were not observed. Other histopathological findings included plasma cell perivascular agglomerates, mitotic figures and perineural mononuclear infiltrates. Finally, all inoculated animals presented differential thickening of the alveolar septum by inflammatory cells in comparison with the lungs of control animals.

Among the variants used, A.2 and Delta resulted in higher histopathological scores, but the Delta variant resulted in greater histopathological changes at 5 and 10 DPI, whereas the peak of the A.2 variant occurred at 5 DPI. The presence of macrophages decreased at 10 DPI in the A.2 variant and remained unchanged in the Delta variant. A detailed evaluation of the parameters used for the score revealed a lower frequency of plasma cells, alveolar oedema and vascular changes in the A.2 variant than in the Delta variant. The Gamma and Omicron variants had lower scores. There was a predominance of lymphocytes in the inflammatory cell population of all of the variants.

In order to synthetize the score system for histopathological findings observed among the distinct experimental groups, we constructed a heatmap (Figure 6) in which each row corresponds to a specific histopathological feature (e.g., alveolar septal thickening, inflammatory infiltrate, vascular alterations), while each column represents an individual animal, systematically grouped according to the SARS-CoV-2 variant and the corresponding day post-infection (DPI).

The colour gradient applied in the heatmap encodes the intensity of each lesion: darker shades of blue indicate higher histopathological scores, while lighter shades correspond to milder alterations, and white represents the absence of a given lesion. A hierarchical clustering method (average linkage with Euclidean distance) was employed to reveal potential patterns of similarity across the dataset.

Hamsters infected with the A.2 and Delta variants exhibited more intense histopathological alterations, particularly between 3 and 10 DPI, whereas animals infected with the Gamma and Omicron variants showed milder and more heterogeneous patterns of pulmonary injury. Furthermore, the temporal distribution of scores suggests a dynamic progression of lung injury, with peak alterations in the acute phase followed by a tendency of resolution by 15 DPI.

### 3.8. Ultrastructural Analysis of the Lungs of Syrian Hamster Lung Samples Infected with the SARS-CoV-2 Variant (Delta Variant or Omicron) via TEM

The profile of the morphological changes observed in the Syrian hamster lung samples was similar regardless of the SARS-CoV-2 variant (Delta AY.6 or Omicron BA.1) and time of infection (3 or 10 DPI). Ultrastructural analysis revealed that thickening of the interalveolar septum resulted in a decrease in the alveolar space (Figure 7E,F), the presence of inflammatory cells and platelets in the capillaries (Figure 7F), vascular congestion (Figure 7C,E,F), haemorrhagic areas (Figure 7D), oedema (Figure 8B), endothelial changes (endotheliitis and cytoplasmic rarefaction) (Figure 7C,D,F), elastin production (Figure 7D), and pneumocyte type II hyperplasia (Figure 8A,B) with mitochondrial vacuolization and disorganisation of mitochondrial crests (Figure 8B). The aforementioned changes were not observed in the animals of the negative control group (Figure 7A,B).

## 4. Discussion

In our comparative study, four different Brazilian circulating SARS-CoV-2 variants (A.2, Gamma, Delta and Omicron) were inoculated into adult male and female *Mesocricetus auratus* at the BSLA-3 facility. These animals were followed for 15 days and presented different phenotypical changes among the variants, which can be useful tools to detect differences between them; however, discrepant findings have been reported by others [23]. These postinoculation changes are highlighted in this manuscript.

Clinically, the hamsters in this study exhibited a non-lethal illness, with no observable clinical signs during our daily inspections. Although similar findings have already been reported, [35] this absence of signs seemed surprising to us, considering the elevated severity of the necroinflammatory lung injury detected in SARS-CoV-2-infected hamsters.

Here, severe inflammatory lung injury was observed during the exudative phase of experimental SARS-CoV-2 infection and was chronologically associated with acute and significant weight loss, mainly in animals inoculated with A.2 and Delta variants, which are also associated with the worst patient outcomes [36].

Gamma and Omicron variants presented nonsignificant weight loss at the exudative step, as confirmed by other authors [37,38]. Since systemic disease was not observed in our study, we suggest that the weight loss observed may be a consequence of anorexia associated with SARS-CoV-2-induced anosmia caused by olfactory bulb injury, which has also been reported in rodents [39,40] and COVID-19 patients [41,42]. Even in Omicron-infected hamsters, the olfactory bulb occasionally shows epithelial cell death [37].

Our histopathological results revealed that Omicron-infected animals had lower overall histopathology scores than other variants did, which was associated with reduced lung injury and inflammation [26,43], which has also been described in patients [20,44,45,46]. Omicron infections in patients are less virulent than those caused by previous VOCs [45], as confirmed in hamsters [21,37,43] and other animal models [47].

The hyperinflammatory state (“cytokine storm”) observed in many patients with severe COVID-19 [48] and/or poor outcomes was not observed in our hamster model. Here, subepithelial mononuclear cell infiltration was the hallmark of SARS-CoV-2-infected Syrian hamster lung parenchyma, which contrasted with the features of acute lung injury (ALI) and the predominance of increased neutrophil influx into the respiratory tract [49]. Our morphological findings pointed to the predominance of innate immunity at 3–5 DPI. The cell populations observed in lung consolidated fields (hepatized areas) included megakaryocytes, eosinophils, macrophages, activated endothelial cells, pneumocytes I and type II pneumocytes with an oxidised form of alarmin HMGB-1, an inflammatory mediator, a classic example of an alarmin [50]. Histologically, the predominance of this cell population suggested that lung injury was mediated by innate immunity, which is similar to DENV infection [51], but in our study, with the absence of two neutrophil markers (myeloperoxidase and Cys-leukotriene) and undetectable C-reactive protein in the hamsters’ blood samples and the absence of inflammatory systemic overflow. In our opinion, the reduced number of neutrophils, unlike the findings described in patients with severe COVID-19, can be explained by the lack of life-support invasive measures and the absence of other respiratory pathogens [52] or by a species-specific immune-mediated response particularly observed in hamsters [53].

Ours SARS-CoV-2-inoculated hamsters presented bronchial epithelial desquamation with cilia loss due to subepithelial mononuclear infiltration with rupture of the lamina propria and submucosa layer with detectable megakaryocytes. This megakaryocyte infiltration was associated with moderate thrombocytopenia. Under physiological conditions, megakaryocytes (MKs) originate from haematopoietic stem cells (HSCs) (resident cells in the bone marrow responsible for megakaryocytopoiesis) and, consequently, produce mature platelets [54]. Similarly, in COVID-19 patients, megakaryocytes and CD61+ platelets in the pulmonary interstitial space are associated with microthrombosis and hyperinflammation in deceased COVID-19 patients [55]. The lung is known to be the primary site where megakaryocyte fragmentation into platelets occurs since it is the first vascular bed encountered after leaving the bone marrow [56]. The loss of lung function as a “platelet manufacturer” after virus-induced endothelial injury confirmed the decrease in the platelet count reported in a hamster model of COVID-19 [57]. We highlight that microthrombosis, NETs, and myeloperoxidase-positive cells were absent in our study; on the other hand, in COVID-19 patients, activated platelets can release vasoactive, haemostatic, and inflammatory mediators, triggering the coagulation cascade, providing a procoagulant surface for secondary haemostasis and, additionally, inducing the release of neutrophil extracellular traps [58]. The resistance to pulmonary microthrombosis observed may be explained by the species-specific mechanism of clot breakdown observed in hamsters during cold-stress hibernation [59]. In our hypothesis, this efficient mechanism of clot breakdown should be studied further until it is well understood since it could be a method of coagulopathy prevention in COVID-19 patients.

The hallmark of the SARS-CoV-2 infection hamster model is respiratory injury, characterised by a mononuclear infiltrate in the interstitial space, which contributes to alveolar septum thickening and subendothelial infiltration, accompanied by damage to the vascular endothelial cell layer (endotheliitis). Parenchymal endothelial damage leads to increased vascular permeability, promoting oedema and respiratory dysfunction, which is a key point in the multifaceted and overlapping processes described for patients with severe COVID-19, leading to a proinflammatory and procoagulant state with the activation of leukocytes and platelets [60]. Notably, in our research, trapped mononuclear cells are present in lymphatic vessels adjacent to and around the circumference of ectatic lymphatic vessels lined with podoplanin-positive endothelial cells, which are also known as bronchial arteries. Podoplanin is involved in lymphatic development [61] and platelet activation since CLEC2 is the only known podoplanin receptor ligand in the context of inflammatory haemostasis [62] and represents an additional risk of venous thromboembolism in COVID-19 patients because many of the systemically released cytokines present a prothrombotic effect in patients [63]. This membrane glycoprotein is commonly found in the endothelium of lymphatic vessels, type I pneumocytes and glomerular podocytes; plays a critical role in lymphangiogenesis and lymphatic development [61]; and influences platelet activation during inflammatory coagulation processes. In our experience, in the hamster model of SARS-CoV-2 infection, classic microthrombosis was not detected in the lung vasculature; only white microthrombi were sparsely detected in the lymphatic vessels, probably resulting from lymphatic stasis and SARS-CoV-2-induced lymphatic endothelial damage, suggesting that lymph obstruction is similar to that described in fatal human COVID-19 patients [64,65]. In addition, the podoplanin-CLEC-2 interaction decreases the contractility of lymph node stromal cells, inhibits the migration of endothelial and lymphatic cells, and contributes to various pathways of the cellular immune response. However, the role of SARS-CoV-2 remains to be better elucidated [66].

Lung lymphatic dysfunction has been described in the pathogenesis of some lung diseases and is associated with increased thrombin and fibrin clots in the lung lymph [67]. Lymphatic clotting has also been described as a morphological finding in severe COVID-19 and is promoted by lymphatic-associated NETosis during patient necropsies [68]. In our hamster model, SARS-CoV-2–associate blood clots were not commonly observed in the lung parenchyma or other organs, unlike in the lungs of patients with severe COVID-19. These results were confirmed by the poor expression of myeloperoxidase and NETosis in the hamster lungs. In our work, we observed two additional compensatory mechanisms of SARS-CoV-2 infection in hamsters. The increased expression of podoplanin in the lungs of SARS-CoV-2-inoculated hamsters may represent a compensatory lymphoproliferative mechanism leading to lymphatic obstruction. Another compensatory mechanism observed was the presence of activated HMBG-1-labelled hyperplastic type II pneumocytes, with abundant cytoplasmic surfactant vesicles and mitochondrial changes in the lung parenchyma, which participate in the maintenance of respiratory homeostasis, as our SARS-CoV-2-inoculated hamsters did not exhibit clinical signs of respiratory distress. This compensatory/reparative respiratory mechanism has already been suggested by other authors [69].

Severe acute lung injury associated with clear histological signals of recovery and the absence of spontaneous death may be explained by the protective mechanism of hamsters against cold lung injury during cool air inhalation since hamsters are winter hibernating rodents. This protective mechanism against lung cold injury is attributed to the cold-inducible RNA-binding protein CIRP- [70,71,72], which reduces wall alveolar thickening and increases airway responsiveness and matrix remodelling [73], promoting rapid recovery from lung injury. CIRP is an RNA-binding protein (RBP) that initially evolved to protect the respiratory tract of hibernating animals against extreme cold (body temperature 37 to 0 °C) [74]; additionally, RBP protects messenger RNA from degradation by endonucleases [75]. Thus, investigating these mechanisms in the pathophysiology of SARS-CoV-2 infection in hamsters is important.

## 5. Conclusions

In this study, we reinforce the robust susceptibility of golden Syrian hamsters to intranasal SARS-CoV-2 inoculation, highlighting phenotypic differences among variants. For example, weight loss and the severity of endothelial injury are dependent on the variant inoculated.

In the hamster, SARS-CoV-2 shows respiratory tropism, characterised by upper and deep respiratory histopathological changes that resemble those of human COVID-19, with strong RNA viral replication in the respiratory tract. Our analysis revealed the predominance of the innate immune response, despite poor pulmonary neutrophilic infiltration in the interstitial space, and an increased presence of eosinophils, megakaryocytes, and lymphocytes, in the absence of a cytokine storm and microthrombosis. Only white microthrombi were detected in the lymphatic vessels. The central role of endothelial injury and its activation, with the presence of proliferative type I and type II pneumocytes with abundant surfactant expression contributing to opened and hyperinflated alveolar fields, as well as the proliferation of intrapulmonary lymphatic vessels, may represent compensatory mechanisms to maintain respiratory homeostasis in golden Syrian hamsters, preventing respiratory distress and death. These compensatory, species-specific pathobiological mechanisms represent new therapeutic options for treating respiratory distress in COVID-19 patients.

## Figures and Tables

**Figure 1 viruses-17-01048-f001:**
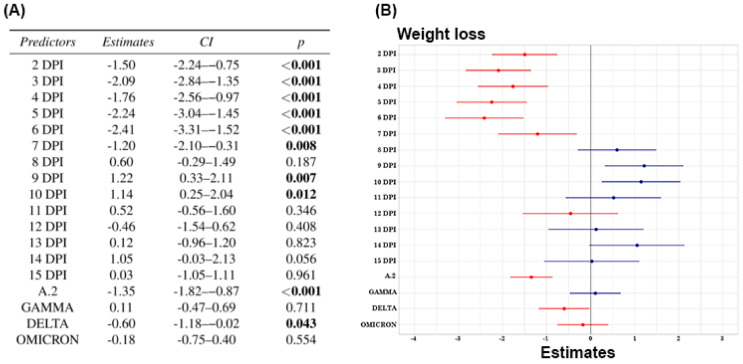
Comparative analysis of body weight loss in hamsters inoculated with four different SARS-CoV-2 variants. (**A**) Analysis of weight loss per day postinoculation and between variants. (**B**) Beta values obtained after applying GLM to the mean weight loss values of the animals, days postinoculation and the four different SARS-CoV-2 variants. The animals were individually recorded. The red lines represent the potential contribution to weight loss, and the blue lines represent the potential contribution to weight gain. *p* < 0.05 was considered statistically significant.

**Figure 2 viruses-17-01048-f002:**
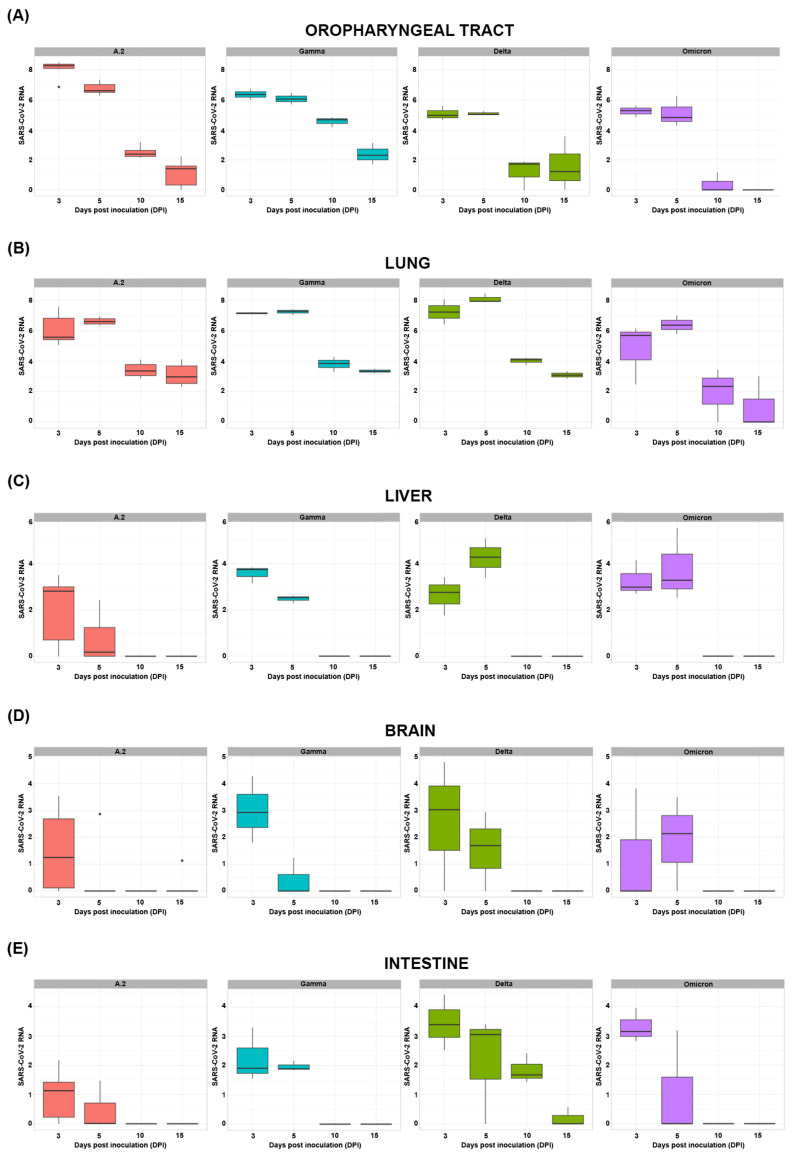
SARS-CoV-2 RNA in the (**A**) oropharyngeal tract; (**B**) lung; (**C**) liver; (**D**) brain; and (**E**) intestine of animals infected with A.2, Gamma, Delta and Omicron variants. The Y-axis represents the SARS-CoV-2 viral load (log_10_ copies/mL). Mock animals presented zero SARS-CoV-2 viral load.

**Figure 3 viruses-17-01048-f003:**
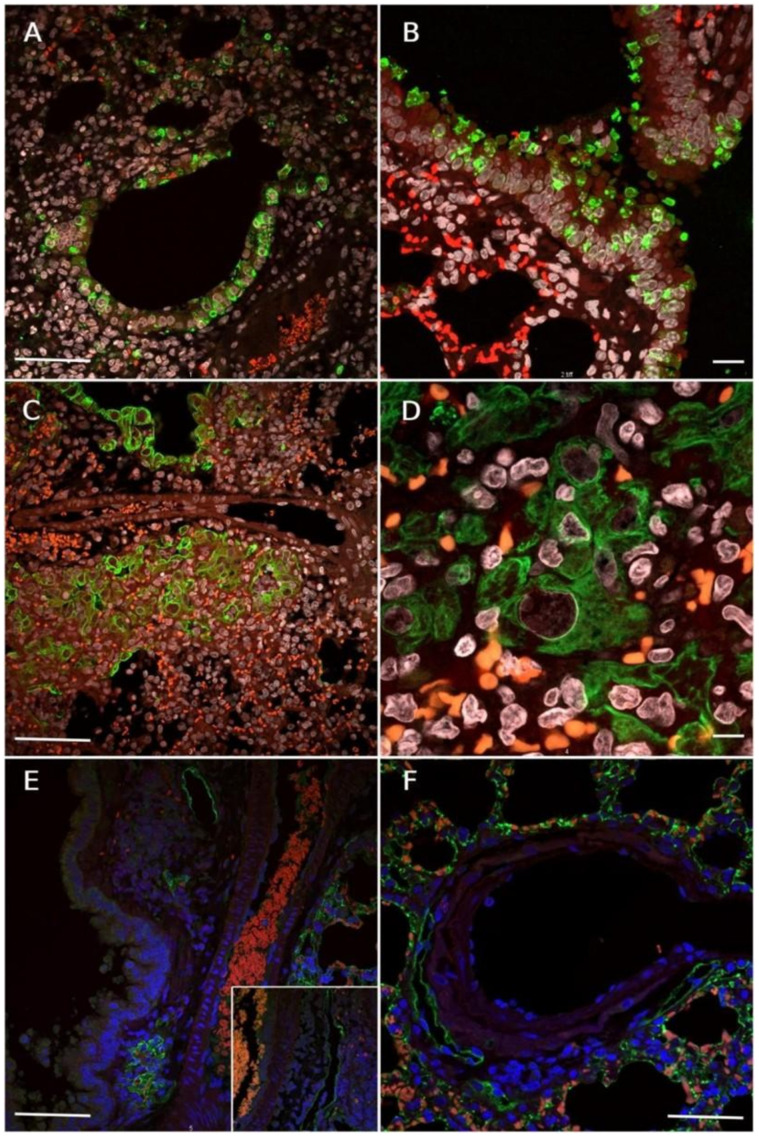
Photomicrographs of the immunofluorescence assay. (**A**) SARS-CoV-2 nucleocapsid in green, showing immunoreactivity for the viral antigen in bronchial cells in marked quantity. Host cell nuclei in blue (DAPI) and erythrocytes in red (Evan’s blue stain). Scale bar = 50 µm. (**B**) In green, SARS-CoV-2 antigens remain in desquamated bronchial cells after evident epithelial hyperplasia. Host cell nuclei in white (DAPI) and erythrocytes in red (Evan’s blue stain). Scale bar = 20 µm. (**C**) Immunolabelling for pan-cytokeratin showing epithelial hyperplasia with atypia (in green). These cells exhibit erratic growth, promoting loss of the usual tissue architecture. Host cell nuclei in white (DAPI) and erythrocytes in red (Evan’s blue stain). Scale bar = 50 µm. (**D**) Immunolabelling for pan-cytokeratin in green, indicating epithelial hyperplasia with atypia. Cellular pleomorphism, anisocytosis, anisokaryosis, evident nucleoli. Host cell nuclei in white (DAPI) and erythrocytes in red (Evan’s blue stain). Scale bar = 10 µm. (**E**) Green podoplanin-positive cells. The lymphatic vessel at the top shows variable amounts of intraluminal leukocytes (magnified picture) and a cluster of type I pneumocytes at the bottom. Host cell nuclei in blue (DAPI) and erythrocytes in red (Evan’s blue stain). Scale bar = 50 µm. (**F**) Lymphatic vessels positive for anti-podoplanin (in green) localised around bronchial arteries. Host cell nuclei in blue (DAPI) and erythrocytes in red (Evan’s blue stain). Scale bar = 50 µm.

**Figure 4 viruses-17-01048-f004:**
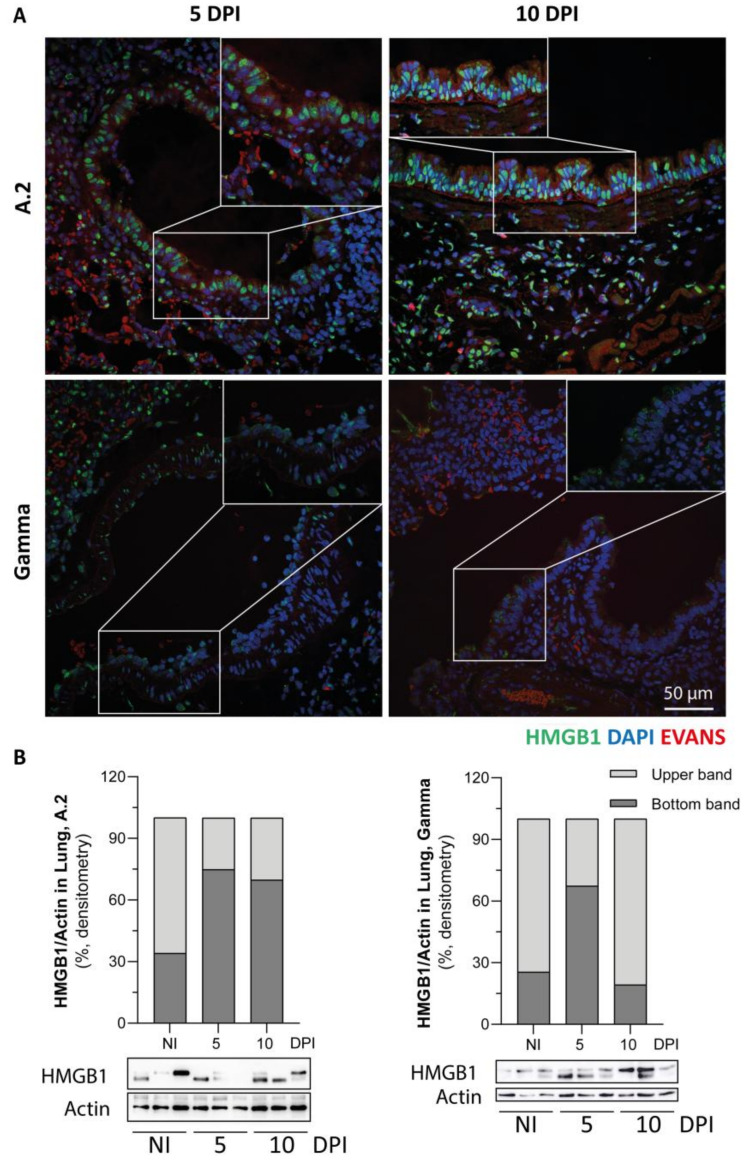
SARS-CoV-2 infection leads to the translocation of HMGB1 to the cytoplasm and induces redox modulation of HMGB1 in the lung. (**A**) Immunofluorescence staining of HMGB1 in the lungs of hamsters infected with A.2 and Gamma variants for 5 and 10 days postinfection (DPI) (scale bar = 50 µm). (**B**) Western blot analysis of HMGB1 in the lungs of noninfected (NI) hamsters and hamsters infected with A.2 and Gamma SARS-CoV-2 variants at 5 and 10 DPI (*n* = 3 per group, one independent experiment using three animals per group), with quantification of bands normalised to β-actin. The upper band corresponds to the reduced form of HMGB1, whereas the lower band represents the disulfide form of HMGB1. The whisker plot refers to the proportion of two bands used for HMGB1 quantification, where the light grey portion corresponds to the upper band and the dark grey portion corresponds to the lower band. Insets, represented by white lines, show magnified areas to better illustrate the cytoplasmic localization of HMGB1.

**Figure 5 viruses-17-01048-f005:**
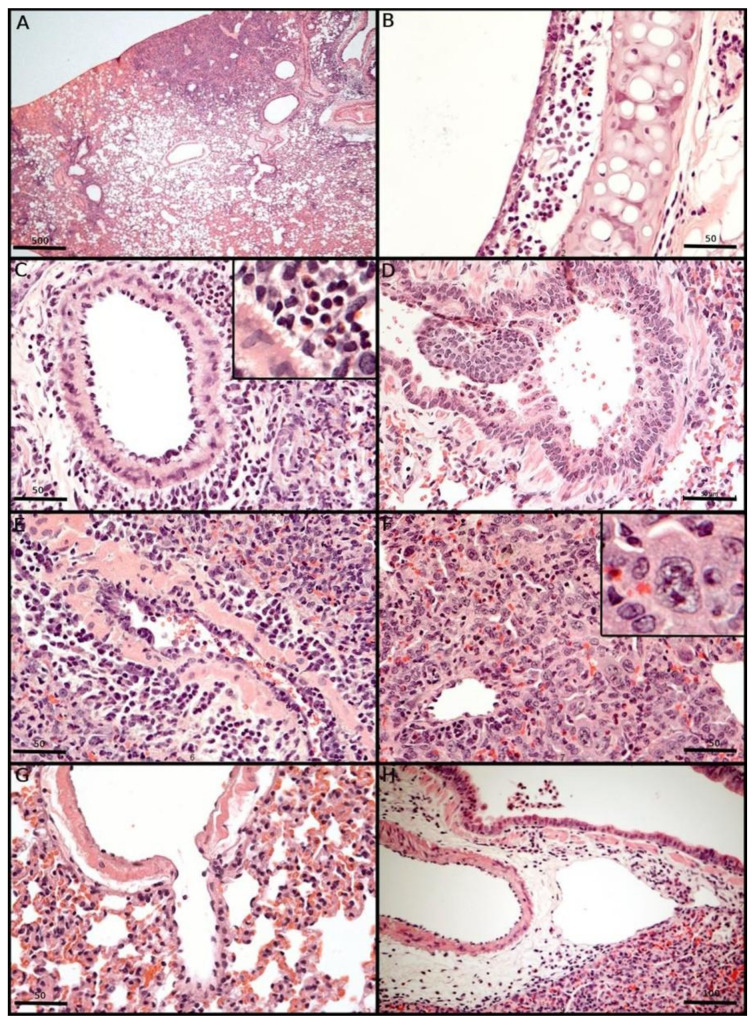
Combined pulmonary pathology severity score summary and photomicrographs of haematoxylin and eosin-stained lung sections from SARS-CoV-2-inoculated hamsters. (**A**): At lower magnification, segmental pneumonia is observed with a heterogeneous interstitial inflammatory infiltrate. Scale bar = 500 µm, HE. (**B**): Loss of the usual morphology of tracheal columnar epithelial cells. There is an absence of cilia on the apical surface of the cells associated with a predominantly mononuclear inflammatory infiltrate between the lamina propria and the cartilaginous rings. Scale bar = 50 µm, HE. (**C**): Perivascular eosinophilic infiltrate. Scale bar = 50 µm, HE. (**D**): Detachment of bronchial epithelial cells associated with intraluminal leukocytes forming so-called “plugs”. Original magnification 400×, HE. Scale bar = 50 µm, HE. (**E**): Severe endotheliitis with detachment of endothelial cells and protrusion into the blood vessel. Scale bar = 50 µm, HE. (**F**): Hyperplasia of epithelial cells with atypia associated with mitotic figures. Scale bar = 50 µm, HE. (**G**): Arterial fistula. Scale bar = 50 µm, HE. (**H**): Marked lymphatic ectasia with focal strangulation of the lymphatic vessel (SETA) caused by the number of inflammatory cells around these structures. Scale bar = 100 µm, HE.

**Figure 6 viruses-17-01048-f006:**
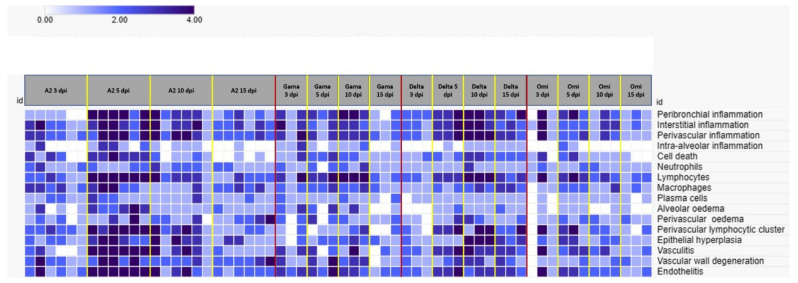
Histopathological changes in the lungs of hamsters infected with SARS-CoV-2, according to semiquantitative analysis by score and represented on the heatmap. The animals were grouped by variants (A.2, Gamma, Delta and Omicron), and within these variants, they were organised into subgroups of days postinfection. The heatmap analyses were carried out using the Morpheus online tool (https://software.broadinstitute.org/morpheus (accessed on 31 July 2024)). Each row represents a histopathological parameter used to apply the scores, and each column represents an animal. The hierarchy average linkage clustering method with Euclidean distance measurement was used. The highest score is represented in dark blue, and the lowest score is represented in light blue. White represents a score of zero for a given parameter.

**Figure 7 viruses-17-01048-f007:**
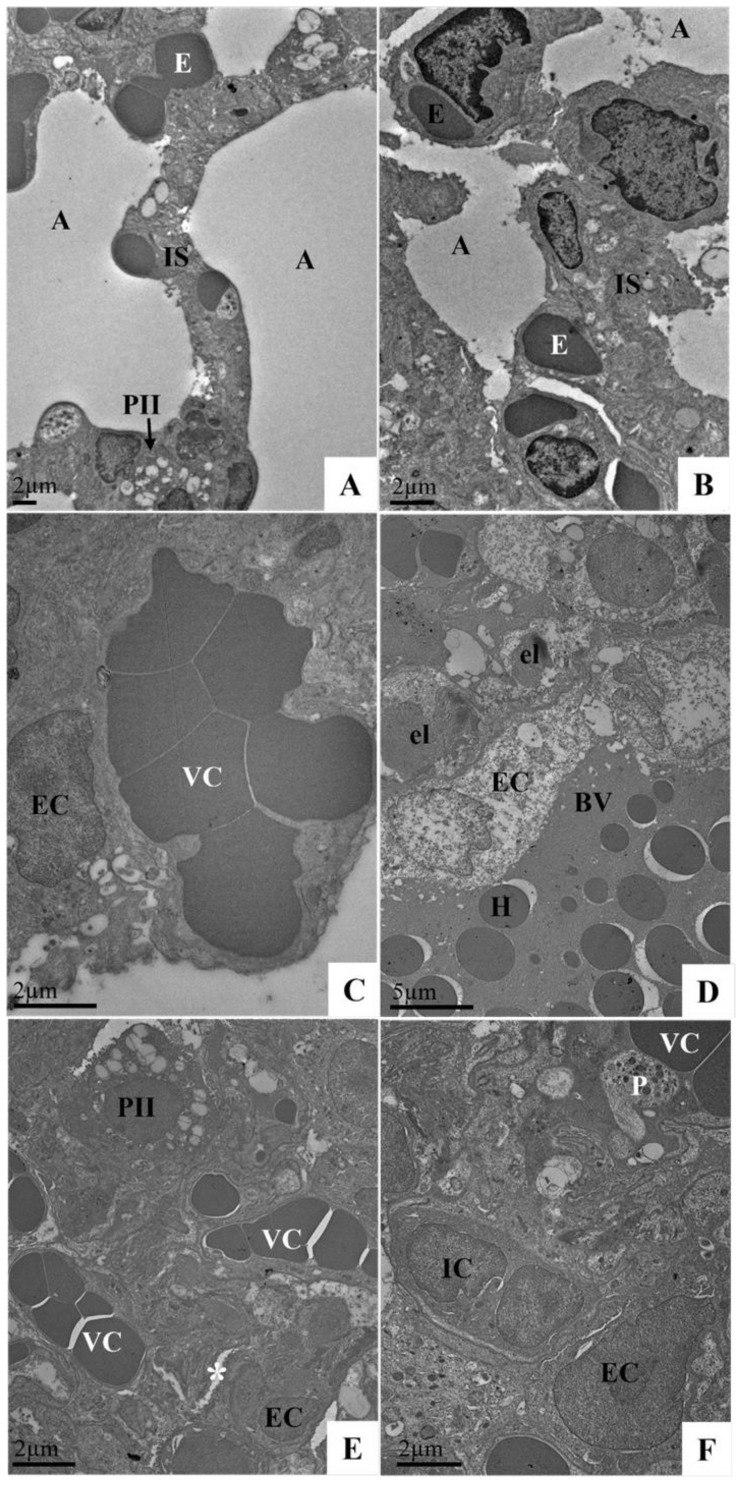
Ultrastructural analysis of Syrian hamster lungs uninfected (**A**,**B**) and infected with SARS-CoV-2 (Delta AY.6 variant) at 3 DPI (**C**) and 10 DPI (**D**–**F**). (**A**,**B**) A: Alveoli; IS: interalveolar septum; PII: Type II pneumocyte; E: erythrocytes. (**C**–**F**) vascular congestion (VC), haemorrhage (H), hyperplasia and cytoplasmic rarefaction of endothelial cells (EC), elastin production (el), type II pneumocyte hyperplasia (PII), a decrease in the alveolar space (*), blood vessels (BV). Erythrocytes (E), inflammatory cells (IC), and platelets (P). Transmission electron microscopy.

**Figure 8 viruses-17-01048-f008:**
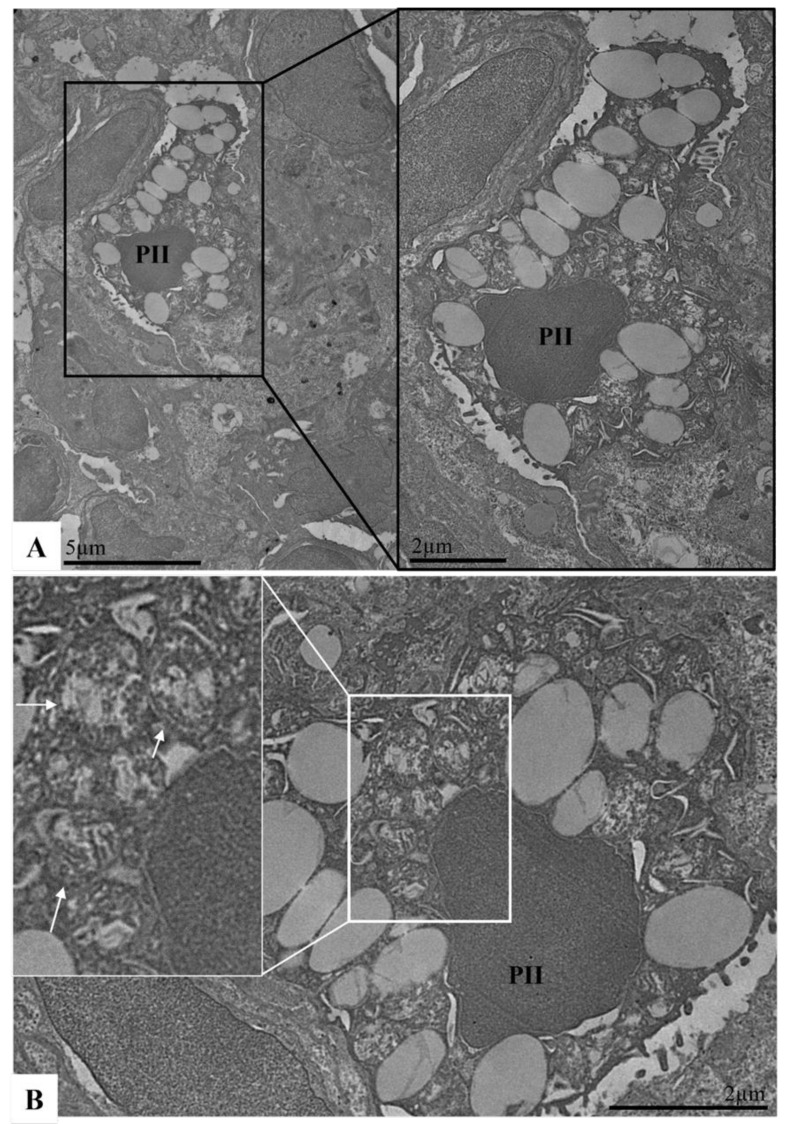
Lungs of Syrian hamster infected with SARS-CoV-2 (Delta AY.6 variant) at 10 DPI. (**A**,**B**) Note: type II pneumocyte hyperplasia (PII) and mitochondrial vacuolization and disorganisation of the mitochondrial crests (arrow). Transmission electron microscopy.

**Table 1 viruses-17-01048-t001:** Average SARS-CoV-2 neutralisation titres * in hamsters inoculated with different strains.

Group	3 DPI	5 DPI	10 DPI	15 DPI
Negative Control	Negative	Negative	Negative	Negative
A.2	Negative	Negative	86.75%	93.08%
Gamma	Negative	30.18%	89.46%	92.87%
Delta	Negative	54.81%	90.42%	95.90%
Omicron	Negative	Negative	60.86%	57.43%

* % means “% Signal Inhibition”.

**Table 2 viruses-17-01048-t002:** GLM analysis * of SARS-CoV-2 RNA content for all organs collected at 5–15 DPI and the variants Gamma, Delta and Omicron.

	Oropharyngeal Tract RNA			Lung RNA			Liver RNA			Brain RNA			Intestine RNA		
Predictors	Estimates	CI	*p*	Estimates	CI	*p*	Estimates	CI	*p*	Estimates	CI	*p*	Estimates	CI	*p*
5 DPI	−0.61	−1.30–0.09	0.086	0.74	0.13–1.35	0.018	−0.32	−1.03–0.39	0.377	−1.02	−1.82–−0.21	0.013	−0.99	−1.60–−0.39	0.001
10 DPI	−4.34	−5.03–−3.65	<0.001	−2.94	−3.55–−2.32	<0.001	−2.71	−3.42–−2.00	<0.001	−1.97	−2.78–−1.17	<0.001	−1.82	−2.43–−1.22	<0.001
15 DPI	−5.33	−6.02–−4.64	<0.001	−3.52	−4.14–−2.91	<0.001	−2.71	−3.42–−2.00	<0.001	−1.90	−2.70–−1.09	<0.001	−2.15	−2.76–−1.55	<0.001
Gamma	0.24	−0.43–0.91	0.480	0.58	−0.01–1.18	0.053	0.82	0.14–1.51	0.019	0.31	−0.46–1.09	0.429	0.71	0.12–1.30	0.018
Delta	−1.37	−2.04–−0.69	<0.001	0.81	0.22–1.40	0.008	1.04	0.35–1.72	0.003	0.50	−0.28–1.28	0.210	1.56	0.97–2.15	<0.001
Omicron	−1.92	−2.59–−1.24	<0.001	−1.28	−1.88–−0.69	<0.001	1.08	0.39–1.76	0.002	0.25	−0.53–1.03	0.533	0.75	0.16–1.33	0.013

* Variant A.2 was established as the reference factor for the Gamma, Delta and Omicron variants and 3 DPI as the reference factor for 5 DPI, 10 DPI and 15 DPI in order to apply the GLM. *p* < 0.05 was considered statistically significant.

## Data Availability

Further inquiries can be directed to the corresponding authors.

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
