# Peer review of "Mesocricetus auratus* (Golden Syrian Hamster) Experimental Model of SARS-CoV-2 Infection Reveals That Lung Injury Is Associated with Phenotypic Differences Between SARS-CoV-2 Variants"

_viruses, 2025, doi:10.3390/v17081048_

Round 1

Reviewer 1 Report

Comments and Suggestions for Authors

This is potentially an interesting paper on the infection of an established animal model with different variants of SARS-CoV-2. It is certainly important to compare and contrast variants in a whole number of ways in animal models and this fits that well (although none of the methods is particularly "innovative", in the sense they are routinely done in these studies, this is not a major problem).

However, there are some major issues that I feel need correcting before this is publishable:

  1. There is no mention in the introduction of any studies that have infected animals (or hamsters) with the variants. For example: Plunkard et al, is only cited once in the discussion. This paper (and others) should be properly introduced and then discussed, highlighting how this study is different or moves the field forward (in any way).
  2. Figure 1 is confusingly presented, at least to me. It appears to be a complex statistical analysis of association. Why not just show the weight loss curves for each infection?
  3. The data in table 1 is not well explained. What does this % mean - does it mean x% of mice seroconverted or x% of virus is inhibited. I appreciate this is a kit and that can be found and referred to, but ideally this detail should be in the paper so that that is not required.
  4. No units or detail is added to the y axes in figure 2. Are these relative amounts compared to mock or absolute numbers? 
  5. Table 2 Is very complex and doesn't really add anything.
  6. Figure 6 is also very complex. Could this be presented in an easier to understand way?
  7. Figures 7, 8 and 9: where are the mock infected controls for this sort of detailed analysis? I am also not sure what 7 and 8 really add, as the histology covers the same points. 
  8. Figure legends are not detailed enough in places. 
  9. Add in "data not shown" when data is not shown. For example in section 3.5. But, in the case of 3.5, I wonder if it is actually important to show these data. 

Author Response

Comments 1: There is no mention in the introduction of any studies that have infected animals (or hamsters) with the variants. For example: Plunkard et al, is only cited once in the discussion. This paper (and others) should be properly introduced and then discussed, highlighting how this study is different or moves the field forward (in any way).

Response 1: Thank you so much for pointing this out. We have accepted your suggestion, adding to the Introduction (line 95) in addition to Plunkard et al, Shiwa-Sudo et al, 2023 (DOI: 10.1128/jvi.01366-22 ), Abdelnabi et al, 2024 (DOI: 10.3390/v16010122) and Wickenhagen et al, 2025 (DOI: 10. 1038/s41467-025-55938-3) to reinforce that, although other research groups have carried out comparative research between SARS-CoV-2 variants in the hamster model, in this work we propose the comparison but in light of the hamster's unique status as a facultative hibernator, central physiological trait to our approach that provides a novel perspective for understanding this host response, which can offer valuable insights for developing strategies to protect human organs, particularly in the context of viral infections. In addition, the entire Introduction has been rewritten (lines 45 to 105) to better present our work.

Comments 2: Figure 1 is confusingly presented, at least to me. It appears to be a complex statistical analysis of association. Why not just show the weight loss curves for each infection?

Response 2: We appreciate your feedback and thank you for these Comments. We have not presented the weight loss curves for each infection due to the limitations imposed by the pandemic, such as the small number of animals available for each group, and Fiocruz adheres to 3R policies for animal experimentation. To carry out a robust and sophisticated statistical analysis and overcome these limitations, we applied the GLM (general linear model), a standard regression model, to analyze the probability of occurrence of the response variable, considering the potential contribution of the explanatory (predictor) variables and the interactions among each variable adopted in our study  The slope of the curve obtained with this mathematical model is the estimate of the β (beta) coefficient, used to predict changes by quantifying the relationship between the predictor variables and the response. In Figure 1, we use Weight ~ dpi + variant to analyze whether the days after inoculation (DPI) and the inoculated variant are significant factors for the weight response variable. The negative estimated β (beta) values, with a confidence interval that does not include zero, indicate the possible contribution to weight loss (response variable) of the predictor variables (DPI and variants) that showed statistically significant p-values. We have included this explanation in the Results section (lines 252-266) to enhance the understanding of Figure 1.

Comments 3: The data in table 1 is not well explained. What does this % mean - does it mean x% of mice seroconverted or x% of virus is inhibited. I appreciate this is a kit and that can be found and referred to, but ideally this detail should be in the paper so that that is not required.

Response 3: Thank you for bringing this issue to our attention. We apologize for the lack of clarity in stating the result. % stands for “% signal inhibition”. We have added this information in line 315. We would like to add that we reviewed the manual for the commercial cPass SARS-CoV-2 Neutralization Antibody Detection Kit (catalog number L00847, GenScript USA Inc./Nanjing GenScript Diagnostics Technology Co., Ltd.) version 7.0, update 2022.02.01, and found that % signal inhibition/antibody neutralization titers above 30% were considered neutralizing for SARS-CoV-2. We proceeded to correct the values in Table 1 and in the text of Sections 2.4 and 3.3. Thus, early production of neutralizing antibodies was only detected at 5 dpi in animals infected with the Gamma and Delta variants (30.18% and 54.81%, respectively). This commercial kit was a single validated option for accessing the total neutralizing antibodies. We regret our failure and thank you for the opportunity to make this correction.

Comments 4: No units or detail is added to the y axes in figure 2. Are these relative amounts compared to mock or absolute numbers? 

Response 4: Thank you for pointing this out. We apologize for our carelessness. These are absolute numbers, the Y-axis represents the viral load of SARS-CoV-2 (log₁₀ copies/mL), the mock animals presented zero SARS-CoV-2 viral load. We have included this information in the legend to Figure 2, lines 336 and 337.

Comments 5: Table 2 Is very complex and doesn't really add anything.

Response 5: Thank you for your critical comment. We decided to append Table 2 to our article as a tool to facilitate the presentation of the probability of occurrence of the response variable (here, the viral load in the tissue) considering the potential contribution of the predictor variables (days after inoculation and inoculated variant), this probability represented in the estimated β (beta) values. Negative β (beta) values, accompanied by a confidence interval that does not include zero and statistically significant p-values, indicate a contribution to the decrease in viral load in the tissue. Positive β (beta) values, accompanied by a confidence interval that does not include zero and statistically significant p-values, indicate a contribution to increasing the viral load in the tissue.

Comments 6: Figure 6 is also very complex. Could this be presented in an easier to understand way?

Response 6: We thank the reviewer for this pertinent observation regarding Figure 6. Indeed, the heatmap presents complex multidimensional data that may be challenging to interpret. To address this, we have expanded the description of the figure in the Results section to clarify its structure and significance.

Comments 7: Figures 7, 8 and 9: where are the mock infected controls for this sort of detailed analysis? I am also not sure what 7 and 8 really add, as the histology covers the same points.

Response 7: Thank you for this question. The mock animals samples were not subjected to electron microscopy, as no alterations were observed under optical microscopy.

Comments 8: Figure legends are not detailed enough in places. 

Response 8: Thank you for pointing this out. We have improved the captions for Figures 2 and 4 (lines 335–337 and lines 401–410, respectively, in the revised manuscript), as well as for Tables 1 (line 315) and 2 (lines 387 and 388).

Comments 9: Add in "data not shown" when data is not shown. For example in section 3.5. But, in the case of 3.5, I wonder if it is actually important to show these data.

Response 9: We deeply appreciate the reviewer’s suggestion and have accepted the inclusion of “data not shown” on line 358. We have retained in the text the information on the results of cys-leukotriene and myeloperoxidase, markers of neutrophils, to corroborate the reduced number of neutrophils in our histopathological results and the undetectable C-reactive protein, which reinforces the absence of systemic inflammation in a hamster model.

Reviewer 2 Report

Comments and Suggestions for Authors

The Mesocricetus auratus (Golden Syrian Hamster) hamsters were inoculated with clinical isolates, expanded and sequenced, that yielded SARS-CoV-2 virus strains A.2, Gamma P.1, Delta AY.6, and Omicron BA.1. The infection model was evaluated clinically, morphologically and virologically over a 15-day time course after infection. The production of neutralizing antibodies over time following infection were also evaluated.  Phenotypic variances were found as well as physiological compensatory mechanisms that maintain respiratory homeostasis that prevents the hamsters from succumbing to severe respiratory distress and death.  The data is convincing and thoroughly and properly analyzed.

However, the manuscript needs improvement.  Firstly, the discussion is too long and disorganized. Secondly, the results are merely descriptive and it is not clear how this research differs from other Covid-19 hamster model studies that are referenced in the paper. The discussion repeats the authors’ results that were obtained in detail to discuss each point. This is not supposed to be.  The description should be only in the results section.

What is unique about this research is not clearly discussed. Why the research was done in the first place should be in the introduction, given the volume of information already existing on SARS-CoV-2 infection of hamsters and other animal models. What characteristics that the authors found have not been reported?  This really needs to be rectified before publication.

In addition, neutralizing antibodies should have been better characterized as IgM, IgA, IgG. The temporal formation of subclasses of Ig may be useful.

How was the oropharyngeal cavity sampled? Was the mouth considered (the oral cavity)?  Please put this in the methods section.  There is little information on the oral cavity in SARS-CoV-2 infection and its immunity.

Minor comments:

Two references are not in the right numerical format:  Plunkard et al., 2023 and Ball et al., 2023 (reference 105?).

Author Response

Comments 1: Firstly, the discussion is too long and disorganized.

Response 1: Thank you for pointing this out to our attention. We sincerely apologize for the lack of clarity in the previous Discussion. This section has been restructured, making it leaner (from line 536 to line 655) with a focus on comparing our results with previous research and highlighting the possibility of future perspectives taking into account the hamster's characteristic of being a facultative hibernator.

Comments 2: Secondly, the results are merely descriptive and it is not clear how this research differs from other Covid-19 hamster model studies that are referenced in the paper.

Response 2: Dear Referee, our reported results represent the gross findings recorded during the study period. The GLM analysis was crucial in interpreting the multiparametric approach employed in our study. In addition, our study was unique in that it compared four different variants and experimentally confirmed the reduced virulence of the omicron variants. 

Comments 3: The discussion repeats the authors’ results that were obtained in detail to discuss each point. This is not supposed to be.  The description should be only in the results section.

Response 3: We sincerely thank the reviewer for the careful reading. We apologize for our carelessness in the previous version of the Discussion section. Some results that were previously included in this section have been moved to the Results section, and the Discussion has been reformulated accordingly.

Comments 4: What is unique about this research is not clearly discussed. Why the research was done in the first place should be in the introduction, given the volume of information already existing on SARS-CoV-2 infection of hamsters and other animal models. What characteristics that the authors found have not been reported?  This really needs to be rectified before publication.

Response 4: Dear Referee, as already described in Comment 2, our manuscript presents robust and significant results after GLM analysis for the first time. Details about changes in lung histology are provided for the first time, including lymphocyte trapping in peribronchial lymphatic vessels with the presence of white clots and the poor presence of granulocytes in necroinflammatory lung injury following SARS-CoV-2 infection. In addition, our study was unique in that it compared four different variants and experimentally confirmed the reduced virulence of the more recent variants. 

Comments 5: In addition, neutralizing antibodies should have been better characterized as IgM, IgA, IgG. The temporal formation of subclasses of Ig may be useful.

Response 5: Thank you for your suggestion concerning our manuscript. Unfortunately, there is a lack of specific reagents adapted for hamsters, such as specific antibodies, which limits our ability to fully assess the temporal formation of Ig subclasses. We recognize that this is a limitation of our study. We have included this observation in the text, referring to Muñoz-Fontela et al. on lines 85-87

Comments 6: How was the oropharyngeal cavity sampled? Was the mouth considered (the oral cavity)?  Please put this in the methods section.  There is little information on the oral cavity in SARS-CoV-2 infection and its immunity.

Response 6: Thank you for your question. The swab was taken from the nasal cavity to the pharynx, not the oral cavity.

Comments 7: Two references are not in the right numerical format: Plunkard et al., 2023 and Ball et al., 2023 (reference 105?).

Response 7: Thank you for your comment. We've made the correction.

Reviewer 3 Report

Comments and Suggestions for Authors

In this manuscript, Rodrigues et al. make an extensive effort to study the influence of several SARS-CoV-2 variants in different biological parameters (generation of neutralizing antibodies, immune cell infiltration, activation of innate responses and histopathology) from Syrian hamsters. The manuscript compiles many interesting results which are worth to be published. However, similar approaches to the present study have already been published, which reduces its novelty component. Below you can find my comments:

Abstract poorly reflects all the work that has been done in this manuscript. For example, there is not even a mention if significant differences in pathology were detected between variants.

Lines 57-78, the information provided here is disorganized and a bit messy. In line 69 it is mentioned that activation of pro-inflammatory mechanisms leads to excess of pro-inflammatory cytokine in blood. This process happens later in a temporal scale when compared to the information provided in lines 76-79, which indicates that activation of innate immune response mechanisms that leads to secretion of IL-6, TNF-alpha and type I interferon.

In line 72, authors make a sudden jump from discussing consequences at the physiological level during cytokine storm syndrome to mention that adaptive immune response is fundamental upon vaccination. These two events are largely different and out of the scope of the main topic. This paragraph (lines 57-78) needs to be improved. 

Line 83, this reads as if transgenic humanized hamster models are available, which is not true. I would make a clear distinction between humanized mice and non-humanized hamster models.

Lines 94-103, in this paragraph authors recapitulate on the advantages of the hamster model. The information provided is clear, although authors should acknowledge the limitation of this model, i.e. low mortality (as referenced by Muñoz-Fontela et al. Nature 2020).

Line 258, I would not say these observations are surprising, many studies before have already reported lack of symptomatology in the hamster model. This is stated in previous studies (i.e. PMID: 39370428, PMID: 37243219)

Lines 264-268, this part of the text should be linked to each part of either Figure 1A or Figure 1B.

Figure 1, this figure is quite difficult to understand considering the information that is provided. What does the ‘Estimates’ parameter mean? Is this the difference between the predicted and the observed weight loss? If so, what is the implication of a negative weight loss? Does this mean that hamsters gained weight? I guess that CI indicates confidence interval, what represent the two columns below this parameter? How many hamsters (n value) were used for each group? All this information should be clarified in the figure caption or main text.

Line 284-285, statements similar to: ‘No differences were found…’ related to the results shown in this manuscript should be linked to the figure representing such data. This should be done for each figure. For instance, in section 3.2 there is no reference to any figure or table, which makes difficult to follow the story.

Figure 3, the elements (upper extreme, box, central line…) that compromise the box and whiskers plots need to be explained. Indicate how many independent experiments were done and what is the number of replicates used for this experiment. Also the headings indicating each variant name have a small font size.

I am concerned about the generation of neutralizing antibodies at 3 DPI. Innate immune responses are dominant during the first 2-4 days of infection, while B cells tend to activate after 3 – 5 days post-infection. IgM might be detectable after 3 days but it is unlikely that this isotype has neutralizing activity. This capacity would come from IgG antibodies which appear as early as 7 days post-infection. How can authors explain the presence of neutralizing antibodies at 3 and 5 DPI? My interpretation is that the rapid test they are using lacks sensitivity, thus this might correspond to false positives. I would suggest to perform more sensitive methods to assess neutralizing activity in sera, as described in PMID 33893470

Table 2, authors explain that variant A.2 and 3 DPI are used as a reference for GLM analysis. Does the data from Gamma, Delta and Omicron variants also to from 3 DPI? Please clarify this information in the table caption.

Lines 347-348, authors claim “There were no differences between the groups for the levels of cys-leukotriene, C-reactive protein and myeloperoxidase”. Where are the results supporting this statement? If this is not shown in the manuscript (which I could not find), please indicate data not shown. Otherwise, cite their corresponding figure/table.

Lines 349, section 3.6, the Immunofluorescence title is very generic and unspecific. Be more specific about what you are analyzing here.

Line 386, leave a space between “cells” and “in”.

Figure 4, results from the western blots (WB) are very inconsistent. First, there is no indication about the molecular size shown in Fig 4B. Also, in the left WB, actin shows two bands while WB on the right side shows a single actin band. I am also concerned about the formation of dsHMGB1 bands in non-infected hamster samples. How can authors explain these results?

The Discussion is too long (around 4 pages). I recommend the authors to make shorter, more direct and simple discussion, restricting themselves on discussing their results instead of introducing new concepts (such as COPD).

Author Response

Comments 1: Abstract poorly reflects all the work that has been done in this manuscript. For example, there is not even a mention if significant differences in pathology were detected between variants.

Response 1: We thank you for the critical comments and helpful suggestions. The Abstract has been changed, even using your suggestion.

Comments 2: Lines 57-78, the information provided here is disorganized and a bit messy. In line 69 it is mentioned that activation of pro-inflammatory mechanisms leads to excess of pro-inflammatory cytokine in blood. This process happens later in a temporal scale when compared to the information provided in lines 76-79, which indicates that activation of innate immune response mechanisms that leads to secretion of IL-6, TNF-alpha and type I interferon. In line 72, authors make a sudden jump from discussing consequences at the physiological level during cytokine storm syndrome to mention that adaptive immune response is fundamental upon vaccination. These two events are largely different and out of the scope of the main topic. This paragraph (lines 57-78) needs to be improved. 

Response 2: We sincerely thank the reviewer for careful reading. The whole Introduction has been restructured. The paragraph in lines 57–78 was indeed confusing, and we have revised it accordingly. We followed the reviewer’s recommendations to maintain the correct temporal sequence and focus on the  information most relevant to this work. Additionally, we included the reference DOI 10.1038/s41577-020-0311-8 to help establish the connection between a dysfunctional innate immune response and COVID-19 outcomes.

Comments 3: Line 83, this reads as if transgenic humanized hamster models are available, which is not true. I would make a clear distinction between humanized mice and non-humanized hamster models.

Response 3: Thank you for pointing this out, we are sorry for our carelessness. We have rewritten the sentence making it clear that these are golden Syrian and Roborovski hamsters and the only transgenic are the humanized mice. This information has been rectified in lines 68 and 69 of the new version of the manuscript.

Comments 4: Lines 94-103, in this paragraph authors recapitulate on the advantages of the hamster model. The information provided is clear, although authors should acknowledge the limitation of this model, i.e. low mortality (as referenced by Muñoz-Fontela et al. Nature 2020).

Response 4: Thank you very much for this suggestion, for encouraging us to write about the limitations of this model. One of the main difficulties we had in carrying out this work was the lack of specific reagents tailored for hamsters, which limited our ability to fully assess the consequences of SARS-CoV-2 infection. We have included this observation in the text, lines 92 to 94 of the new manuscript, referencing Muñoz-Fontela et al. Nature 2020. The absence of mortality among SARS-CoV-2-inoculated hamsters, in our interpretation, represents a good health condition of the animals. Additionally, the experimental design adopted did not address this question; therefore, additional experimental time and the absence of euthanasia over time would be necessary. 

Comments 5: Line 258, I would not say these observations are surprising, many studies before have already reported lack of symptomatology in the hamster model. This is stated in previous studies (i.e. PMID: 39370428, PMID: 37243219)

Response 5: Dear Referee, our surprise was justified by the elevated severity of the necroinflammatory lung injury detected in SARS-CoV-2-infected hamsters. We are confident that respiratory homeostatic, species-specific mechanisms contributed to the absence of gross clinical signs. Nevertheless, we have revised this passage in the new version of the manuscript and cited DOI:10.3390/v15051133 as one of our references in line 545 of the new version, clarifying that the absence of clinical signs in SARS-CoV-2-inoculated hamsters is not unprecedented.

Comments 6: Lines 264-268, this part of the text should be linked to each part of either Figure 1A or Figure 1B.

Response 6: We deeply appreciate the reviewer's suggestion. We have proceeded to include the requested explanation in the text. In lines 252 to 266 of the new version of the manuscript.

Comments 7: Figure 1, this figure is quite difficult to understand considering the information that is provided. What does the ‘Estimates’ parameter mean? Is this the difference between the predicted and the observed weight loss? If so, what is the implication of a negative weight loss? Does this mean that hamsters gained weight? I guess that CI indicates confidence interval, what represent the two columns below this parameter? How many hamsters (n value) were used for each group? All this information should be clarified in the figure caption or main text.

Response 7: Thank you for bringing this to our attention We sincerely apologize for the lack of clarity in the outcome statement. The estimate β (beta) coefficient is the slope of the curve obtained from the general linear model (GLM). It is used to predict alterations by quantifying the relationship between the predictor variables and the biological response. It is a robust and sophisticated statistical analysis, based on interactions among multiple variables adopted in our study and on 3R policies implemented by Fiocruz. Moreover, it facilitated circumventing limitations imposed by the pandemic, such as the reduced and declining number of animals in each group, as detailed in the Materials and Methods section.

In Figure 1 we use Weight ~ dpi + variant to analyze whether days post inoculation (DPI) and the inoculated variant are significant factors for the weight response variable. Negative estimated β (beta) values, with a confidence interval that does not include zero, indicate a potential contribution to weight loss (response variable) of the predictor variables (DPI and variants) that showed statistically significant p-values.

Thus, the data shown in Figure 1 indicates that the animals lose weight significantly depending on the days after inoculation and the variant inoculated, where the interaction of the period between +2 and +7 dpi and the variants A.2 and Delta showed a statistically significant factor for the average weight loss.

We clarified these aspects in the main text, lines 252 to 266 of the new manuscript.

Comments 8: Line 284-285, statements similar to: ‘No differences were found…’ related to the results shown in this manuscript should be linked to the figure representing such data. This should be done for each figure. For instance, in section 3.2 there is no reference to any figure or table, which makes difficult to follow the story.

Response 8: Thank you for this comment concerning our manuscript. We did not include a figure for the biochemical and hematological parameters analyzed because, after statistical analysis, most of these parameters did not show differences between groups. However, we have reported the most relevant findings in the manuscript as part of the description of the hamster model presented here.

Comments 9: Figure 3, the elements (upper extreme, box, central line…) that compromise the box and whiskers plots need to be explained. Indicate how many independent experiments were done and what is the number of replicates used for this experiment. Also the headings indicating each variant name have a small font size.

Response 9: Thank you for your comment. We are considering Figure 4 to respond to you. We have described the whisker plot in the figure legend (lines 401-410), and we clarified that this evaluation was performed only once, using three different animals per group (line 405). The variant name is clearly indicated in each experimental approach, and we did not make any changes to it.

Comments 10: I am concerned about the generation of neutralizing antibodies at 3 DPI. Innate immune responses are dominant during the first 2-4 days of infection, while B cells tend to activate after 3 – 5 days post-infection. IgM might be detectable after 3 days but it is unlikely that this isotype has neutralizing activity. This capacity would come from IgG antibodies which appear as early as 7 days post-infection. How can authors explain the presence of neutralizing antibodies at 3 and 5 DPI? My interpretation is that the rapid test they are using lacks sensitivity, thus this might correspond to false positives. I would suggest to perform more sensitive methods to assess neutralizing activity in sera, as described in PMID 33893470

Response 10: We sincerely thank the reviewer for his careful reading. When we reviewed the manual for the commercial cPass SARS-CoV-2 Neutralization Antibody Detection Kit (catalog number L00847, GenScript USA Inc./Nanjing GenScript Diagnostics Technology Co., Ltd.) version 7.0, update 2022.02.01, we found that % signal inhibition/antibody neutralization titers above 30% were considered neutralizing for SARS-CoV-2. We proceeded to correct the values in Table 1 and in the text of Sections 2.4 and 3.3. Thus, early production of neutralizing antibodies was only detected at 5 dpi in animals infected with the Gamma and Delta variants (30.18% and 54.81%, respectively). The Plate Reduction Neutralization Test (PRNT) was not standardized at our institution during the period in which these experiments were carried out, and the serum from the hamsters described here ran out when the other serological tests were carried out. We recognize that this is a limitation of our study.

Comments 11: Table 2, authors explain that variant A.2 and 3 DPI are used as a reference for GLM analysis. Does the data from Gamma, Delta and Omicron variants also to from 3 DPI? Please clarify this information in the table caption.

Response 11: We appreciate your critical comment. To apply the GLM  analysis the probability of occurrence of the response variable (here, the viral load in each tissue analysed) considering the potential contribution of each of the predictor variables (days after inoculation and each inoculated variant), it was necessary to establish variant A.2 as the reference factor for the Gamma, Delta and Omicron variants and 3 DPI as the reference factor for 5 DPI, 10 DPI and 15 DPI. We have included the requested information in the table caption (lines 387-388).

Comments 12: Lines 347-348, authors claim “There were no differences between the groups for the levels of cys-leukotriene, C-reactive protein and myeloperoxidase”. Where are the results supporting this statement? If this is not shown in the manuscript (which I could not find), please indicate data not shown. Otherwise, cite their corresponding figure/table.

Response 12: We deeply appreciate the reviewer’s suggestion. Following the recommendation, we have decided to include the expression 'data not shown' in the revised manuscript (line 358).

Comments 13: Lines 349, section 3.6, the Immunofluorescence title is very generic and unspecific. Be more specific about what you are analyzing here.

Response 13: Thank you for pointing this out. We have proceeded with the change: “Immunofluorescence reaction for SARS-CoV-2 nucleocapsid, lymphatic vessel endothelial marker (podoplanin) and HMGB1 translocation in lungs of SARS-CoV-2 inoculated hamsters” (lines 359 and 360).

Comments 14: Line 386, leave a space between “cells” and “in”.

Response 14: We sincerely thank the reviewer for careful reading. We made this correction (line 391).

Comments 15: Figure 4, results from the western blots (WB) are very inconsistent. First, there is no indication about the molecular size shown in Fig 4B. Also, in the left WB, actin shows two bands while WB on the right side shows a single actin band. I am also concerned about the formation of dsHMGB1 bands in non-infected hamster samples. How can authors explain these results?

Response 15: Thank you for the comment. HMGB1 is a highly mobile protein in SDS-PAGE, typically migrating at ~25–30 kDa depending on its glycosylation status. Under non-reducing conditions, an additional lower band may appear below the main band, corresponding to the disulfide form of HMGB1 (ds-HMGB1). The detection of ds-HMGB1 indicates that oxidative stress—often driven by inflammation—can lead to HMGB1 oxidation, and this oxidized form contributes to the amplification of the inflammatory response.

We have now included the raw Western blot (WB) image along with the molecular weight ladder to confirm the size of the bands in the Non-published Material file. Please note that the membrane was cut before imaging. This step was necessary because both α-HMGB1 and α-actin antibodies were incubated on the same blot. Since actin is highly abundant, its signal was overexposed during development, which compromised the detection of HMGB1. Therefore, the membrane was cut and developed separately to improve band clarity.

Interestingly, the appearance of ds-HMGB1 in the non-infected (NI) hamster also surprised us. We believe this particular animal may have a specific susceptibility that resulted in the presence of ds-HMGB1.

Regarding the two bands observed in the actin WB, this is likely due to overexpression, leading to non-specific bands. This is also evident in the raw WB image included in the Non-published Material file.

Comments 16: The Discussion is too long (around 4 pages). I recommend the authors to make shorter, more direct and simple discussion, restricting themselves on discussing their results instead of introducing new concepts (such as COPD).

Response 16: We sincerely thank the reviewer for the careful reading. The Discussion has been reformulated accordingly, focusing on discussing our results and highlighting some future perspectives.

Round 2

Reviewer 1 Report

Comments and Suggestions for Authors

The authors have responded well to all of my previous comments and I thank them for that.

I remain concerned with 3 minor areas:

  1. The introduction does not describe the findings from other studies in this model. Which is needed for understanding how this study fits in.
  2. The lack of controls for figure 9 is concerning, but perhaps the authors could highlight a normal mitochondria to show the difference upon viral infection.
  3. The electron microscopy in figures 7 and 8 basically repeats the histology data, so does not add a lot. I would not insist on its removal - but I think it is worth considering. Again, lack of controls makes these data tricky (the whole point is that mocks will look normal, so will highlight the effect seen - I fully expect them to be "normal"). 

Author Response

Comments 1: The introduction does not describe the findings from other studies in this model. Which is needed for understanding how this study fits in.

Response 1: We sincerely thank the reviewer for the valuable and critical comment. We have highlighted the general findings from other studies in lines 94–99 of the new version of the manuscript.

Comments 2: The lack of controls for figure 9 is concerning, but perhaps the authors could highlight a normal mitochondria to show the difference upon viral infection.

Comments 3: The electron microscopy in figures 7 and 8 basically repeats the histology data, so does not add a lot. I would not insist on its removal - but I think it is worth considering. Again, lack of controls makes these data tricky (the whole point is that mocks will look normal, so will highlight the effect seen - I fully expect them to be "normal"). 

Response 2 and 3: We thank the reviewer for the valuable comments. The electron microscopy figures have been restructured accordingly. Although the electron microscopy data partially overlap with the histological findings, we believe that the ultrastructural analysis provides a higher level of detail, thereby strengthening and supporting the histopathological results. For this reason, we have chosen to retain these figures. To address the concern regarding the lack of controls, we have included representative images from uninfected control animals in the revised figures.

Reviewer 2 Report

Comments and Suggestions for Authors

The authors have adequately addressed all of the comments and critiques of the first review.  The discussion is much improved.

Author Response

The authors have adequately addressed all of the comments and critiques of the first review.  The discussion is much improved.

Response 1: We sincerely thank the reviewer for the careful reading of our manuscript, as well as for the critical comments and helpful suggestions. We are pleased to know that the discussion is now clearer and improved. Your feedback was truly helpful in improving our work.  

Reviewer 3 Report

Comments and Suggestions for Authors

All my concerns have been addressed properly. Authors have made a significant amount of changes to improve the quality of their manuscript. Thus, I endorse this manuscript for publication.

Author Response

All my concerns have been addressed properly. Authors have made a significant amount of changes to improve the quality of their manuscript. Thus, I endorse this manuscript for publication.

Response 1: We sincerely thank the reviewer for the careful reading of our manuscript and for the encouraging comments. We are glad to know that the changes we made helped improve the quality of the work. Your feedback was truly appreciated and motivating.